# Creating the Multifaith Chapel, 1938–1955: Architecture and the Changing Understanding of "Religion"

Jeanne Halgren Kilde

Program in Religious Studies, College of Liberal Arts, University of Minnesota, Minneapolis, MN 55455, USA; jkilde@umn.edu

**Abstract:** Interfaith or multifaith chapels are so ubiquitous now in the United States—present in colleges and universities, hospitals, shopping malls, and airports—that their development as a distinct architectural form is often taken for granted. Yet that development in the mid-twentieth century was complex and even fraught. Taking a religious studies approach, this article examines the development of three early examples—the Chapel of the Four Chaplains, the Brandeis University chapels, and the MIT Chapel—to reveal the gradual movement, conceptual and architectural, toward a viable space serving many religions. While the former two examples proved unsuccessful in their goal of establishing a shared interfaith space due to their reliance on an understanding of religion as discrete traditions that resulted in exclusivist incompatibilities, the latter example moved beyond the emphasis on traditions to advance an unconventional, phenomenological understanding of religion as individual experience and spiritual life, and by doing so successfully achieved the goal of creating a space amenable to practitioners of many traditions, or none. Further, this article demonstrates how architecture functioned as a constitutive component in the developmental and popularization of this fresh understanding of religion and religious experience.

**Keywords:** MIT chapel; Chapel of the Four Chaplains; Brandeis University; Eero Saarinen; phenomenology; spirituality; architecture; tri-faith America; Matthew Nowicki; interfaith space

## 1. Introduction

Multifaith chapels are ubiquitous in contemporary religious and public landscapes. College and university campuses, hospitals, and other institutions have long completed the process of transforming their former religiously specific chapels into spaces welcoming to all religious and spiritual persuasions, and public places such as airports and shopping malls routinely provide such spaces as well. The spaces are given various names—non-denominational, interfaith, or multifaith chapels;[1] spiritual centers; meditation or reflection rooms; and so forth—but they all share the intention of accommodating practitioners of all religions, or none. The development of these spaces as a distinct architectural form is often taken for granted, with architectural specialists and the general public alike viewing them as natural, commonsense responses to an increasingly multicultural society. But a closer historical examination of such spaces reveals not only that they are actually a relatively new phenomenon but that their evolution was far more complex and less certain or preordained than commonly thought; in fact, it was often highly fraught and included at least a couple of high-profile false starts before the creation of what may well be the first successful example of the form, certainly the earliest publicly touted example, architect Eero Saarinen's multifaith chapel at MIT, which opened in 1955.

How did this novel form of religious space develop? What forces motivated those who propelled its development? And what issues were at stake? How was architectural design involved? And how did the creation of this new type of space affect how "religion" itself, as an important category of human behavior, was understood? Examining the mid-twentieth-century history of the development of interfaith religious spaces in the United

States through three specific architectural efforts, this essay reveals that a revolution in how the concept of religion was understood was required to overcome social, political, and religious difficulties involved in their creation. As we will see, while early calls for interfaith space were based on established understandings of religion as discrete traditions that rendered shared space untenable, this novel building type successfully emerged only with the acceptance of an unconventional understanding of religion, one that perceived it not in terms of traditions but as individual experience. Moreover, each of these three cases demonstrates how architecture functioned as a constitutive component in the developmental process and, most importantly, in the popularization of the fresh understanding of religion and religious experience that informed the successful effort at MIT.

## 2. Religious Exclusivism vs. Tri-Faith Pluralism: The Chapel of the Four Chaplains in Philadelphia and the Brandeis University Chapel

The first two efforts we will examine emerged from the mid-twentieth century notion that America was a tri-faith nation composed of Protestants, Catholics, and Jews—what I call a "new pluralism" emphasis—and both were tripped up by long-embedded notions of religious exclusivism. The tri-faith notion emerged as a direct counter to the long history of religious separation and exclusiveness predicated on Western theological perspectives that viewed religious traditions as necessarily in conflict with one another as each attempted to assert or defend their own particular beliefs and truth claims.[2] This exclusivist view extended not only across the Christian/Jewish divide, resulting in an entrenched, structural antisemitism, but also across Christian groups as well, illustrated in Catholic/Protestant antipathies that had for decades fueled anti-Catholic episodes across the United States and among and within the many Protestantism denominations with long histories of external disputes and internal schisms.[3] This traditional view was beginning to be challenged in the first half of the 20th century, and the process was accelerated during World War II as combat situations brought Christians of all stripes and Jews together just as military chaplaincies were professionalizing, resulting in government-sponsored efforts to accommodate the range of religious preferences.[4] What arose in the military as an instrumental need to accommodate diverse religious groups in limited spaces soon migrated into the civilian sphere.

### 2.1. The Chapel of the Four Chaplains in Philadelphia

On the home front, a story of military bravery propelled a harrowing scene of tri-faith patriotism and self-sacrifice into the public arena, fostering a growing public awareness of the possibility of tolerance and cooperation among the three religions and significantly broadening acceptance of the new pluralistic view. The story of the Chapel of the Four Chaplains began with the deaths of four US Army chaplains—two Protestant ministers, a Catholic priest, and a Jewish rabbi—who were posthumously awarded the Distinguished Service Cross for heroism and sacrifice related to their actions in February 1943 when an American troopship, the *S.S. Dorchester*, on which they were being transported along with hundreds of troops, was attacked and sunk in the North Atlantic by German U-boats. As the sinking ship was being abandoned, the four clergymen handed out life vests to the troops and, as the supply dwindled, ultimately gave their own vests to troops before they linked arms and prayed on the ship's deck as it went down. All four perished. When the award was made, a year after the event occurred, it was reported in newspapers and magazines around the country, and the heroic story of self-sacrifice was widely seen as symbolic of the ideals of patriotic cooperation and unity of purpose across faiths that characterized American society (or should do so) and contrasting starkly with the exclusivism, intolerance, and Old-Worldism prejudices of Europe.[5]

The effort to create a tri-faith chapel came about in 1946, when the Reverend Daniel Poling, the father of one of the Protestant chaplains aboard the *Dorchester*, launched a campaign to memorialize the four heroes with an interfaith chapel that would accommodate Protestant, Catholic, and Jewish services. The pluralist, tri-faith model was central to Poling's conception

of the chapel, and he deployed it as an effective fund-raising tool, inviting the then-notorious Rat Pack entertainers—Frank Sinatra, Sammy Davis Jr., and Peter Lawford, themselves two Catholics and a Protestant, respectively—to perform at fund-raising events.[6] Donations arrived from various sources, including the American Legion, the Kresge Foundation, and B'nai B'rith, confirming the effort as multifaith (Weart 1951). See also, (The Story of the Four Chaplains Through Original World War II Documents n.d.).

To create the chapel, Poling repurposed the lower level of the church he pastored, the Baptist Temple in Philadelphia. He filled the 30′ × 60′ longitudinal space with several rows of chairs split by a center aisle and focused on a raised stone platform or bema at one end. This platform was three-sided and could be rotated to display one of three altars and backdrops depending upon the type of service being conducted: a Protestant one with a table on which were placed a cross and candlesticks, a Jewish bema with an *aron hakodesh* topped with the Ten Commandments and an eternal light hanging above, and a Catholic altar with a crucifix above on the back wall[7] (Figure 1). A stairway leading down to the chapel displayed a painting of the four clergymen and a plaque identifying the chapel as an "interfaith memorial" and a "sanctuary for brotherhood".

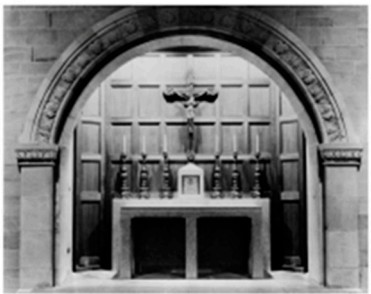

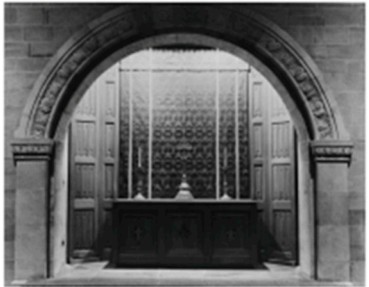

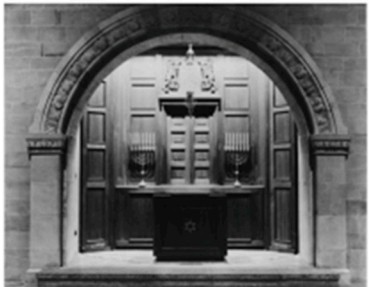

**Figure 1.** Catholic, Protestant, and Jewish altars in the Chapel of the Four Chaplains. Courtesy The Chapel of the Four Chaplains, Philadelphia, PA https://fourchaplains.org/wp-content/uploads/2020/03/Four-Chaplain-Poster-1.pdf (accessed on 28 January 2024).

The dedication ceremony in February 1951 characterized the chapel as a triumph of American religious unity and positioned the building's significance within the growing Cold War political situation. The featured speaker was President Harry S. Truman, who interpreted the memorial as a testament to the American ideal of "unity" across Protestantism, Catholicism, and Judaism. In his words, "the unity of our country is a unity under God. It is a unity in Freedom, for the service of God is perfect freedom." In this speech, the chapel itself served as a robust rejoinder to anti-religious Soviet ideology as Truman

extended the meaning and importance of the chapel beyond religious cooperation among the three faiths to that of a new, national, politico-religious identity (Truman n.d.). See also, (Herzog 2011, pp. 77–81) (Figure 2).

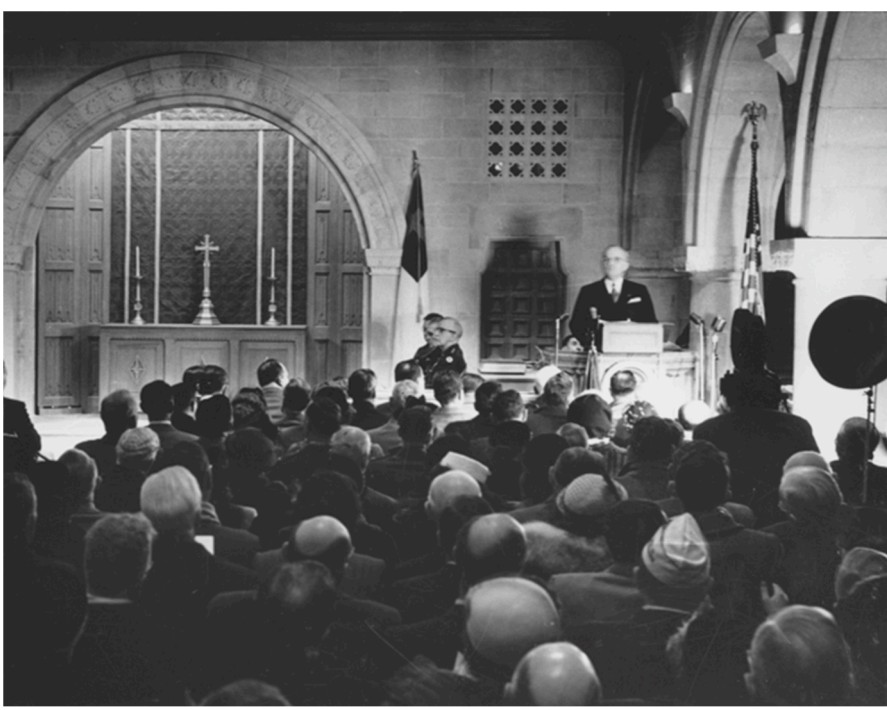

**Figure 2.** "President Harry S. Truman speaks at the Chapel of the Four Chaplains on the eighth anniversary of the sinking of the S.S. Dorchester," 3 February 1951. Courtesy of the Special Collections Research Center, Temple University Libraries, Philadelphia, PA.

This national identity, fusing one understanding of religion (that of tri-faith religion) with loyalty to the nation, rested on a belief in the divine foundations of the American national culture, a belief that was the central premise of what historian Jonathan Herzog calls the "spiritual-industrial complex." (Herzog 2011, pp. 5–6). This Christian nationalist ideology, growing since the early twentieth century, was boosted significantly by this and other situations during the Second World War. As historian Kevin Schultz observes, proponents of this view felt that "democracy could not survive without a deeply felt religious faith, especially one premised on individual human dignity." (Schultz 2011, p. 45). This faith, grounded in "a personal God of righteousness and love, sacredness of all men as children of a common Heavenly Father, and a realization of God's will by practicing social justice and brotherhood" was understood to be diametrically opposed to Hitler, constituting the well-spring of the "American way of life." (Schultz 2011, p. 45). As the Cold War developed in the late 1940s, governmental and social institutions embraced religion as a counter-ideology to Communism, and American advocacy of individual human dignity was construed as directly opposed to the anti-religious policies of the Soviet Union and to the Communist embrace of historical materialism. The Four Chaplains story and the tri-faith identity of the nation demonstrated the superiority of American democracy and egalitarianism over the godless materialism of Communism. (Herzog 2011, pp. 5–6). In this ideological context, religion took on a new, public role and character, fusing religious and civic commitment.

Although Truman and others were adamant about a unity across religious traditions, the tri-faith ideal was fragile, and two situations that bookended the formation of the chapel—one occurring during its development and the other several years later—exposed its precarity and, at the same time, demonstrated the power of the new public religion/patriotism matrix. Problems surfaced early in the development of the Chapel



of the Four Chaplains when the Archbishop of Philadelphia, Cardinal Dennis Dougherty, publicly expressed his disapproval of the project and refused to allow the chapel's altar to be consecrated or the space to be used for Catholic services. (Weart 1951); (Four Chaplains 1944). Poling went ahead, nevertheless, and included the Catholic altar in honor of Chaplain John P. Washington, who perished aboard the *Dorchester*. For an early fund-raising dinner in 1947, Poling sought three speakers to represent each tradition and invited the new congressman from Massachusetts, John F. Kennedy, to represent Catholic participation. Kennedy initially agreed to speak at the dinner, but shortly before the event, he sent his regrets, saying he was not comfortable appearing as a spokesperson for the Catholic faith. Stung by this last-minute decision, Poling accused Kennedy of backing out under orders from Archbishop Dougherty. Kennedy responded that he had never discussed the event with the archbishop and that when he learned he was to "represent" Catholics as spokesperson, he felt compelled to decline because he had no clerical training or credentials to attend in that capacity.[8] The dinner went on without Kennedy and, indeed, the chapel opened in 1951, but for Poling the episode remained a sore spot, and he raised it again in his 1959 autobiography, long after the chapel had been completed and dedicated. Echoing the anti-Catholic sentiments of what historian John McGreevey has termed "cosmopolitan intellectuals" who claimed that Catholicism was not compatible with democracy, Poling recounted the episode, saying that Kennedy's refusal to appear at an event to honor these national heroes demonstrated that his loyalty to the Catholic hierarchy superseded his loyalty to the nation and thus threw into question his suitability for public office.[9] In Poling's words, "...at least once John Kennedy of Massachusetts submitted, apparently against his own inclinations and better judgment, to its [the Roman Catholic Church's] dictates." (Poling 1959, p. 261).

Poling's memoir came out just as Kennedy was preparing to run for the US presidency, so the stakes were high as it fueled both doubt about Kennedy's suitability and anti-Catholic sentiment, but the Four Chaplains incident also opened an opportunity for Kennedy to directly address the issue of a Catholic's suitability for office. In advance of a Kennedy campaign event in Houston, Texas, Poling coached a close friend and fellow Protestant minister who would be attending Kennedy's address to the Houston Ministerial Association to interrogate Kennedy about the reasons he declined the invitation to attend the Four Chaplains fund-raiser in 1947 and to press him on the role of the archbishop in the decision. Kennedy's response, which followed up on his answer to an earlier question during the event, took a spatial turn, focusing on where Catholics worshipped. In the earlier statement, he had affirmed that as a Catholic he could attend services in a Protestant church—"private ceremonies, weddings, funeral and so on". In response to the specific question about the chapel fund-raiser, he explained that in Poling's invitation, the Four Chaplain's chapel had been characterized as an "interfaith chapel", but he later learned it was not truly so as it was "located in the basement of another church", and because of this "there had never been a service of my church because of the physical location". In this situation, he could come only as "a citizen", not as a spokesman for the Catholic faith at dinner to raise funds "when the whole Catholic church group in Philadelphia was not participating and because the chapel had never been blessed or consecrated."[10] Thus, in his view, the Chapel of the Four Chaplains was *not* an interfaith chapel.

But what exactly *was* an "interfaith chapel"? A designated space in the basement of a Protestant church would simply not do for many Catholics in Philadelphia, a view that is not surprising in a city where nativism had been rampant a little over a century earlier and where, more recently, the purveyors of "cosmopolitan liberalism" had targeted the city's Polish Catholics as an example of an insular Catholic culture at odds with the democratic American culture.[11] Religious exclusivism remained strong in the city, with Protestants and Catholics eyeing one another with deep suspicion.[12] Thus, the effort to create a physical space dedicated to religious pluralism exposed the frailty of the Protestant/Catholic/Jewish conception. The Chapel of the Four Chaplains was "interfaith" on Protestant terms, physically and conceptually enveloped by the Baptist church. With their Protestant "host",

the other two faiths did not occupy equal positions; instead, the relationship may well have seemed more like the notion of toleration of religious others that had obtained in an earlier era.[13] In effect, the Chapel of the Four Chaplains contravened the goal of serving as an interfaith religious space accommodating the pluralistic, tri-faith understanding of American society, and exposed the reality that the three faiths were far from equal.

## 2.2. Interfaith Space at Brandeis University

A second effort to create a new pluralist chapel that would serve Protestants, Catholics, and Jews together under one roof occurred at Brandeis University in the early 1950s. This effort arose from the young institution's effort to serve its multifaith student body, which reflected the tri-faith model of the American populace, by providing a "non-sectarian" context for the proposed chapel. Criticism of the effort, however, fueled by exclusivist and essentialist understandings of religious traditions, prompted a rethinking of the project that left the initial shared-space concept on the drawing board and replaced it with three separate chapels.

Founded in 1948 as a Jewish-sponsored, nonsectarian university, Brandeis University developed as a direct response to the quota systems used by many colleges and universities to limit the number of Jewish students enrolled. All would be welcome at Brandeis. The institution was in the propitious position of being able to create a new campus on its Waltham, Massachusetts, site, albeit in the midst of several extant buildings from the medical college that previously owned the land, so in 1950 they hired the architectural firm of Saarinen, Saarinen and Associates (SSA) to develop a master plan for the campus that would include an interfaith chapel. As articulated in the presentation book that architects Eero Saarinen and Matthew Nowicki submitted to college leaders that same year, Brandeis was committed to the "non-denominational pattern of instruction [that] is one of the precious traditions of American education", but the book noted in the section on the chapel, that did not mean it would be "indifferent to the religious life or students" nor would it "minimize[s] the wholesome significance of the religious experience". Introducing a generalized language of the spiritual, the description continued, "fact and data must be integrated with value and purpose, else the student is left without spiritual anchorage."[14] The book went on to express the architects' understanding of the role of the interfaith chapel, echoing elements of the tri-faith idea and emphasizing the need to move beyond mere tolerance to equality:

> The chapel obviously must provide a place of worship that will be used in common by students of Catholic, Protestant, and Jewish persuasion. It must serve to emphasize the equality of all creeds rather than the pseudo-liberalism of a "least common-denominator" tolerance. The spirit of this approach is reflected in the architectural design for an interfaith chapel which faithfully mirrors the University's nonsectarian principle while preserving the integrity of each form of religious worship. (Saarinen, Saarinen, and Associates n.d., p. 31)

Nowicki, a recently immigrated Polish architect and friend of Saarinen's whom he invited to collaborate on the project, created an expressionistic design for the chapel that addressed this goal by proposing a centralized gathering space focused on a three-sided, rotating altar similar to the one that would be included a year later in the Chapel of the Four Chaplains (Figure 3). As with the Four Chaplains space, this one would also accommodate all three religious groups at different times. This spatial conception garnered regional attention when an article in the *New York Herald Tribune* declared, ". . . reflecting in architectural design, the university's nonsectarian principle, [it] will preserve each form of religious worship." (Brandeis U Plans 50 New Buildings 1950). Locally, the plan was announced in the Brandeis student newspaper and received strong student support.[15] Nevertheless, the design was soon abandoned, and in March 1951, Saarinen submitted an alternate plan that eliminated the revolving altar and reorganized a triangular gathering space into three separate wings each with its own altar or bema (Figure 4). Like the initial design with its single space and rotating altar, this one also attempted to represent the

equality of the three traditions spatially under a single roof, but Brandeis leaders found it problematic because it required participants in any service to be seated with their backs to the other groups' altar/bema, which may have been interpreted as a sign of disrespect.[16] By 1953 this design and the entire attempt to create an interfaith building with a unified space to be used equally by each faith was abandoned, and rather than a single building, the institution would erect first a Jewish chapel and later additional, separate chapels for Catholics and Protestants, a course of action that, as we will see, did not sit well with many students.[17]

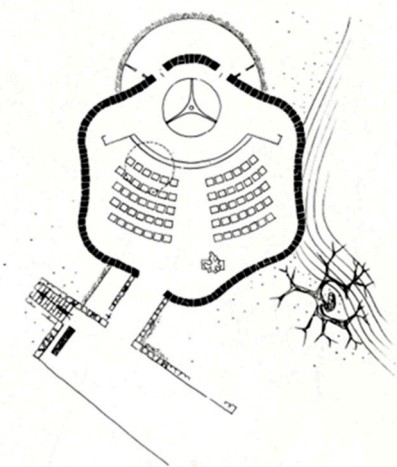

**Figure 3. Matthew** Nowicki's plan for Brandeis Chapel. Courtesy of Cranbrook Archives, Cranbrook Center for Collections and Research.

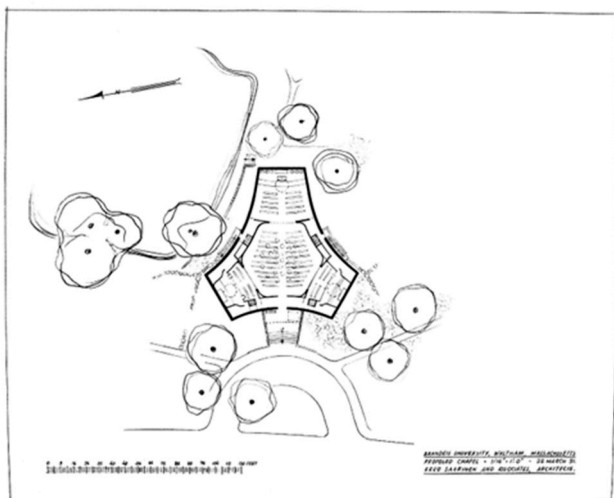

**Figure 4.** Eero Saarinen's second design for Brandeis Chapel. Courtesy of Cranbrook Archives, Cranbrook Center for Collections and Research.

The reasons for rejecting both Nowicki's and Saarinen's designs exposed an additional nuance in the difficulty of creating a religious space that could effectively serve three faiths and their worship gatherings. Brandeis President Abram L. Sachar articulated an exclusivist and essentialist understanding of religion as he announced the decision: "You can't dilute in a religious edifice. A chapel barren of religious symbols becomes bereft of any significance." (New Chapel Will be Jewish Sponsored—And Sectarian 1953, p. 1). Years later, as he reflected on the decisions to first abandon the rotating altar and later to abandon the single chapel concept, Sachar reiterated this exclusivist concern, noting the altar's "gimmickry", but citing as more important the concern that the proposed space would produce "no feeling that this was one's own house of worship, that it belonged emotionally

as well as physically to the group using it." (Sachar 1976, p. 70). Religious practices across traditions were simply not compatible; in fact, they were mutually exclusive.

Certainly, this concern stemmed in part from the question of Catholic participation in a shared-space chapel. The editorial staff of the student newspaper alleged that the administration's rejection of the initial designs was based on an essentialist understanding of Catholic worship: "because of the Catholic tradition which does not permit a Catholic student to worship in an edifice which serves a denomination other than his own," but they questioned whether the issue had been thoroughly researched and urged the administration to pursue more discussion with the diocese on the subject. (New Chapel Will be Jewish Sponsored—And Sectarian 1953, p. 1). The Brandeis Student Council along with several other students also objected, having taken Nowicki's conception of non-sectarian spatial equality to heart. They expressed concern that erecting a Jewish chapel contradicted the nonsectarian character of the institution, which was established to counter the Christian sectarianism of other institutions of higher education that undergirded restrictions on Jewish enrollments. They asserted that a Jewish-identified chapel erected first and built larger than the Catholic and Protestant ones would necessarily be seen as accentuating the institution's identity as hegemonically Jewish and thus contradict both the ideal of tri-faith equality and the institution's nonsectarian commitment. The students further contended that they had not been consulted on the abandoning of the original plan for a single interfaith chapel, and they charged, again, that the question of Catholic participation in a shared building, which had been articulated initially as the main obstacle, needed more research.[18]

Although some practical issues also informed the decision, including the early failure of fund-raising efforts for the Protestant and Catholic chapels, the principle in question involved the nature of religious pluralism and relationships among the three faiths. In the language of one editorial in the Brandeis student newspaper, it was "a question of values," specifically, "whether the encouragement of religious expression should consist in an emphasis on religious differences or an emphasis on the common aspirations of all religions." (Editorial, A Question of Values 1953, p. 2). While the Student Council, newspaper, and others advocated for the latter, the traditional practice of religious separation championed by the administration carried weight with other students, who argued that any space that did not express difference implicitly diluted religion and identity and was therefore anti-religious. As one Christian student wrote to the newspaper, "to the devoutly religious person, regardless of creed, the very idea of an interfaith chapel is laughable...To strip it of all meaning, is to arrive at a "meditation room"...This is not non-sectarianism, this is irreligion." (Zadig 1953).

The latter view prevailed. The resulting three buildings, designed by the firm of Harrison and Abramovitz, would ultimately communicate a separate-but-equal relationship among the traditions. The three chapels, each one a modernist sculptural statement reportedly intended to "resemble a partly open bible," were all of the same height and positioned so that none would cast a shadow on either of the other two. However, the Jewish chapel, seating 125, was larger than the others (which seat 60) in order to accommodate the greater number of students likely to use it. The buildings faced an outdoor "interdenominational area" to be "used when common purposes were to be served." (Sachar 1976, p. 70). In this way, religious exclusivism was maintained, but the possibility of spatial cooperation was also made available. A report on the dedication of the buildings by *Life Magazine* heralded the achievement as "Three Faiths in Harmony", and pronounced the buildings "separate but related". A sense of identity and mutual engagement was emphasized by Sachar who was quoted as saying: "Here at Brandeis, please God, we shall each respect our own faith and carry this respect with pride in the presence of each other." (Three Faiths in Harmony 1955, p. 113).

Thus, even though the nonsectarian admissions policy of the campus intended to accommodate students of differing religious backgrounds and each of the three traditions had its own organization (Jewish Hillel, the Catholic Newman Club, and the Student

Christian Association), creating a worship space that all three groups would share proved controversial, for just as with the Chapel of the Four Chaplains, the visible ownership of the space by a specific group, Protestants in its case and Jews in the Brandeis case, was seen as necessarily relegating the other traditions to a secondary status that resulted in a sense of visitor-ship rather than ownership for the non-host groups and thus gainsaid the collective equality of all three. By autumn 1955, when the up-coming dedication of the new Brandeis chapels was reported widely by the national press, the three buildings were hailed as an innovative solution to the interfaith space question that was increasingly plaguing college campuses. (Gordon 1955; Three Brandeis Chapels will be Dedicated 1955).

One additional situation may also have influenced the course of action taken by the Brandeis leadership, particularly in light of issues that had arisen in Philadelphia. Not everyone was happy with the three chapels solution; in particular, a conservative ex-priest from Boston, Leonard Feeney, who had been stirring anti-Jewish and anti-Protestant sentiment among his followers for years, was livid. Acting, in his view, as a defender of the true Catholic faith, he attacked Brandeis for building a Catholic chapel and excoriated Boston Archbishop Richard Cushing for agreeing to hold Mass in the building at its dedication, a decision that in Feeney's book evidenced either the archbishop's ignorance or his malice.[19] Feeney wailed, "a Catholic Archbishop has been persuaded to place the One True Faith, the Mass, and the Holy Eucharist, on a par with heretical perversions and even with Jewish perfidy," and he urged the Catholic subscribers to his newsletter to respond: "Anxiously, we ask the prayers of our readers that somehow, by some unforeseeable intervention, this plan will be frustrated, and that our Jesus in the Blessed Sacrament will be spared the desecration of dwelling in sanctuary on the campus of Brandeis, as the tenant and the target of the Jews." (Feeney 1955; Feldberg 2012, pp. 112–13)[20] Feeney's followers rose to his call, distributing antisemitic flyers that targeted the campus and threatened violence; in response, Brandeis organized police protection for the dedication services. (Feldberg 2012, p. 113).

Feeney's antisemitic hate speech, like the grilling of John F. Kennedy by the Houston Ministerial Association, demonstrates the persistence of the exclusivist and separatist understandings of religions and the high stakes that were at play in efforts to bring religious groups closer together in physical spaces. Brandeis accomplished a unique solution, bringing together Protestants, Catholics, and Jews in a shared outdoor space between the buildings while at the same time providing separate indoor worship spaces. Building ownership, assertions of religious identity, and the representation of equality were maintained in a fragile indoor-outdoor balance within the ostensibly "non-sectarian" context of the broader campus itself.

The efforts of the Chapel of the Four Chaplains organizers and Brandeis University to create chapels that accommodated the new pluralist, tri-faith notion of the national character constituted a distinctive shift in thinking about the character and function of religious space as well as significant progress toward the development of a new kind of space. Neither institution, however, achieved the goal of creating an interfaith space. A contemporaneous effort to create such a space did succeed, however, in large measure because its developers shifted the discourse surrounding their chapel in a very different direction.

## 3. Re-Conceptualizing Religion at MIT

While the Four Chaplains and Brandeis University projects were launched to address ethical concerns of equality and cooperation among the three religious traditions qua traditions and to accommodate the worship of each in a unified space, the successful interfaith chapel erected on the campus of the Massachusetts Institute of Technology (MIT) in 1955 emphasized concerns related to phenomenology, metaphysics, and anti-materialism that resulted in a unique conviction that the most important aspect of "religion" was not identification with a particular tradition or engagement in a particular worship practice, but the widely shared human experience of "spirituality".

*3.1. An Early Chapel Plan for MIT: William Welles Bosworth and the "Imponderables"*

Interest in creating a chapel on the campus of the Massachusetts Institute of Technology initially surfaced not in the post-war period with the attendant tri-faith idea but in the 1930s with a Protestant-based, latitudinarian desire to encourage students' spiritual life that arose out of changing cultural circumstances. Responding to the rise of spiritual free thinking and atheism among college students, and keenly aware of the all-encompassing materialism of the scientific and technical focus of the institute's mission and curriculum, the dean of MIT's Department of Business and Engineering, Erwin Schell, teamed with architect William Welles Bosworth, who had recently completed designing the new campus, to lobby for the creation of a chapel for the use of students and others in the campus community.[21] As Bosworth wrote to Schell, "Certainly this is a most propitious moment in the history of the world, when the contrasts of Christ and Anti-Christ are so evident in international ethics today, for the chapel question to be agitated."[22]

Although both men were Christian, their vision of religion was unconventional, calling for a new architectural form. The architectural plans that Bosworth sent to Schell in 1938 conceived of a building that, although Christian in its neo-classical form and style, acknowledged the growing interest in spirituality and atheism, and was called "a chapel or meditation hall."[23] The entablature above the entry, he insisted, should have an inscription that would bring to mind the "importance of the 'imponderables' in the development of man," and he suggested the text be something like "dedicated to the unfolding spiritual life of man."[24] While the textual message would proclaim a generalized ideal of spiritual life, sculptural figures of the prophets and angels (Michael, Gabriel) would mark the building as within the Judeo-Christian realm. The Greek cross plan of the interior space would be surmounted by a double dome, with an oculus in the lower portion allowing sight of a ring of angels (harkening to William Blake's morning star), clouds, and a dove, "emblematic", Bosworth wrote, quoting from the Book of Genesis, "of the Spirit ascending and descending upon the Son of Man. . .." The lighting would be indirect and the walls reminiscent of the mosaics in Ravenna's St. Vitale Basilica.[25] Despite the characterization of the building as a "meditation hall", the spatial organization of the interior followed that of auditorium churches with curved pews arranged on the sloping floor to facilitate hearing "inspiring words and music."[26] Galleries all around would provide additional seating, and choir stalls were positioned in front with a pulpit on the left pier and a reading desk near the right.

Despite these worship accommodations, the idea was to also accommodate individual reflection, and after describing the church-like features, Bosworth launched into a lengthy discussion of more unique elements included for the benefit of more progressive, liberal religionists, pointing to the double-meanings, theological and metaphysical, embedded in the building:

> When one meditates, it is of course a great help to look at something inspiring. The altars of cathedrals express this uplift of thought, and there should be an inspiring note at the visual center of this interior at the spot commonly occupied by an altar. I recommend that it should take somewhat the form, with an elongated marble table, but instead of candles and cross or tabernacle, which would offend the 'free thinker', I should have a great vertical glass panel illuminated by some throbbing, pulsating, electrical currents, which no one knows better how to produce than the experts at Technology, and which would express, it seems to me, to anyone, the presence and the activity of the creative energy and life that has made us and of which we are a part.[27]

This art glass representation of transcendent energy would visually capture the metaphysics of the day, but its articulation in the hands of even this liberal Protestant artist would be secondary to divinity. Bosworth continued:

> On the altar and lit from below, I should place two angels in sculptured glass. They should be kneeling, in an attitude of prayer, on either side of a newborn babe, also sculptured in glass and lit from below. This babe would not necessarily

mean the Christchild, but the birth of a man—any man—which is the most mysterious, the most inspiring, the most glorious thing we know and the thing which should plunge us into deepest thought. Lit from below, in glass, these things would create a beautiful climax such as only light can accomplish to the eye, and with the softly moving light of the electricity in the glass panel on the wall back of them, I cannot conceive of anything more inspiring to look at. The glass should have a slightly golden hue, for warmth and color, disappearing into blue at the top.[28]

In this passage, the double meaning is again offered. Are the figures a celebration of Christianity or the universal humanism of free-thinking perspectives? Theology or metaphysics?

Lastly, Bosworth addressed the color scheme, emphasizing the use of natural, organic tones throughout:

...the green of the walls for the quieting and restful quality of green, like the green leaves of the forest. Gold is the most intellectual color and leads to thoughts and comfort. The brown of the furniture and the floor, which should be carpeted (for both warmth and acoustics), suggesting the brown of the tree trunks and the ground tone, will make a harmony of color appropriate to the usage of the building, and psychologically correct.[29]

Bosworth's description is an eloquent material articulation of the affinity between metaphysical thought, transcendentalism, science, and liberal theology. Within such a space MIT students and faculty, from freethinker to Christian traditionalist, might all feel comfortable. Humanism, meditation practice, the natural environment, spiritual reflection, and Christianity are blended comfortably into a pleasing whole in this early conception of a religious space equally accommodating to a range of spiritual and even secular perspectives.[30]

Bosworth's chapel conception seems to have aligned with Schell's, but despite their efforts, this chapel was never to be built.[31] Another decade would pass before the desire to build a chapel on the campus resurfaced and came to fruition. Yet Bosworth's vision would not be entirely discarded, for as a chapel project developed in the 1950s, some of his ideas would again surface.

*3.2. The Cold War, Personal Religion, and Spirit at MIT*

By 1950, the national ethos aligning patriotism, democracy, and religious faith, driven by Cold War hostility to Communism and its materialist/atheist ethic, pushed MIT leaders to reconsider the idea of building a chapel. As a secular institution dedicated to science, MIT found itself in jeopardy of being identified with the wrong side of this oppositional equation, and in response, advocacy of some form of religion emerged as a means of demonstrating its American loyalty and claim to the moral high ground despite its materialistic mission.

That materialist mission had risen to new heights by mid-century with MIT's engagement in the war effort. By the end of WWII, it was the leading US institution advancing "big science" for military purposes, receiving over USD 100 million in government contracts annually by 1960. Scientific research at MIT achieved advances in radiation, microwave, and other technologies that were quickly put to use by the US military, resulting in the institution becoming the largest non-industrial defense contractor in the country. (Leslie 1993, pp. 6, 11–12, 14–23).[32] MIT's growing leadership in the development of what would later be termed the military-industrial complex coupled with growing national commitment to religion as a counter to Communist materialism brought new purpose to the efforts of like-minded liberal Christians lobbying the institution to support students' religious development alongside their technical education. Dean Schell, for one, sought to temper the scientific, materialist emphasis of the curriculum by providing students with opportunities for humanistic, ethical, and religious reflection. The unspoken intention was that the

presence of religious activity on campus could mitigate potential criticism of the institution as being hyper-materialist and thus atheistic.[33]

Two incidents in the late 1940s made public the institution's awkward position vis-à-vis the growing Cold War rhetoric and provided Schell and others with reasons to revive the need for a chapel. First, in 1945, a national security situation occurred when a Nazi effort to infiltrate the United States by landing spies on the Eastern seaboard was uncovered and the investigation led to the arrest and conviction of a former MIT student, William Colepaugh, on espionage charges. Described as an antisemite, Colepaugh had defected to Germany after dropping out of MIT. As the story of Colepaugh unfolded in the national newspapers, Schell pointed to it as evidence of the need for the institution to address the moral and religious needs of students and, more specifically, the need for a chapel. Referring to an earlier idea to combine an auditorium with a chapel, Schell wrote to MIT president, Karl Compton:

> The arrest of the saboteurs in Maine and the fact that one of them had been an M.I.T. student made me reflect on our chapel project and the need of moral instruction at M.I.T. You probably saw Dr. Butler's recent address to Columbia students in which he stressed his belief that the future of civilization depends on the <u>moral</u> instruction of the youth of today.[34]

The implication that MIT had failed in its obligations to its students and to democratic society raised the stakes on the importance of building a chapel.[35]

It was in the discipline of economics, however, where a second threat to the institution's reputation proved even more troubling. The incident arose in 1947 when a preliminary draft of an economics textbook by MIT faculty member Paul A. Samuelson was judged by a member of the institution's governing board to be insufficiently supportive of laissez-faire capitalism and perhaps even socialistic—and therefore anti-America—in its advocacy of government interventions in the economic system. Through the next two years, President Compton and his successor James Killian defended Samuelson and the publication of the book, which came out in 1950. Reviews split along political lines with conservatives reading it as inappropriately critical of capitalism. A particularly damning evaluation appeared the next year in William F. Buckley's widely read book, *God and Man at Yale*, which reproached liberal universities for having weak moral and religious values and took the discipline of economics, in particular, to task, explicitly naming Samuelson's textbook as an example of how institutions of higher education were advocating socialist ideas (Giraud 2014, pp. 139–43, 146–47, and passim); (Buckley 1951, pp. 49, 71–81, and passim). See also, (Weintraub 2014, pp. 53–54). This was not the kind of publicity the institution wanted.

With such incidents in the public square in the late 1940s, discussions about religion at MIT re-emerged and included a return to ideas that had been percolating among campus leaders a decade earlier. In 1947, just as the textbook controversy was brewing, Schell edited a collection of essays that included one by then-president Compton, titled "Why Religion?" which argued that religion and science were well aligned with and even improved by one another. Referencing psychologist Henry C. Link's popular 1937 book *The Return to Religion*, which emphasized an understanding of religion that highlighted personal religion, service to others, rationality, and compatibility with science, Compton averred that Christianity's admirable core message of love and equality under God was obfuscated by rituals, authorities, creeds, and dogmas, but that the superficiality and irrelevance of these trappings were now, fortunately, being unmasked by "scientific methods of thinking." He wrote, "it is perhaps, one of the great indirect results of modern science that there is gradually developing more of a spirit of religious tolerance in which, outside of certain fundamental religious concepts or attitudes, the details of creed and doctrine are considered more in terms of individual preference than as boundary lines between damnation and salvation." Moreover, he argued, there was a strong need "for religious forms adapted to various personalities." Religion, in his view, was a matter of personal choice: "I believe, the form and expression of a man's proper religious life are those which give him the greatest

spiritual comfort and inspiration toward the better life." (Compton 1947, pp. 92–93, 98). See also, (Link 1937).

Geared to a society embracing scientific secularism, this concept of *personal religion*, a notion that scholars would later term "religious individualism", was gaining appeal, particularly among theologically liberal Christians and Jews, as an alternative to the more ideologically conservative tri-faith notion.[36] MIT's Dean of Students, Unitarian minister Everett Moore Baker, shared the outlook, reiterating the idea of personal religion in a 1949 address: "Each of us will find his own religion and his own way to celebrate his religion—some in loneliness pondering the great and ultimate mysteries, some in quiet communion with others in clean, white meeting houses, some in great cathedrals where color, music, and incense motivate their aspirations." For Baker, this "religion" provided a "sense of belonging to Something bigger than ourselves—and a pattern and method of relating ourselves to that Something." (Baker 1951b, p. 104). Moreover, in his estimation, the "religious" character of the MIT student had more to do with commitment and action toward the improvement of human life and securing human rights. This religion, he argued, takes its inspiration more from "humanistic [rather]than theistic" endeavors, and, he continued, "its methods are in their opinion scientific." (Baker 1951a, pp. 122–23). In Baker's conception, the metaphysical, the humanistic, and the scientific merged within the personal religion of the student.

As James R. Killian took up the office of the president of the institution in 1948, Baker encouraged him to establish a chapel that would serve students engaged in personal, religious reflection—a place where "an individual could go for a moment of quiet meditation". Baker argued that the chapel should not be a building to be used for services but for "small discussion groups or meeting of students of varied denominational interests and connections". He described what we would now call an inter- or multifaith approach, with the chapel serving "the Intervarsity Christian Fellowship, the Hillel Association, and other such organizations" and "symboliz[ing] the religious interests and aspirations of our total community rather than any particular traditional religious group within it", and he speculated that "it might prove to be a very interesting educational experiment".[37] Killian responded that there were no funds for a chapel and Baker should find a suitable room in an extant building, but he would soon become a champion of the effort, particularly as the concerns about religious belief, American loyalty, materialism, and atheism outlined above deepened within the scientific community. A year later, the depth of interest in addressing religion at MIT and elsewhere was driven home during an international convocation on the Social Implications of Scientific Progress that was held at MIT in April 1949, when the segment that proved to be of most interest among the scientists, scholars, students, and guests in attendance was the one that focused on "Science, Materialism, and the Human Spirit", and the panel addressing the specific topic of "Science and Faith" attracted a standing-room-only audience. (Burchard 1950, pp. 196–251).

In addition to these growing concerns about religion expressed by administrators and the scientific faculty, appeals to establish a chapel were coming from other quarters as well. In November 1949, several Catholic student associations at MIT, having been encouraged by a recent statement by Killian acknowledging that "the Development Committee realizes the need of the spiritual development of the students and that there are tentative plans for a chapel at M.I.T.", sent their own request for regular Catholic services and a chapel to hold them in. Echoing the earlier objections to the Chapel of the Four Chaplains and the Brandeis Chapel, the Catholic students noted that a "non-sectarian" chapel might serve the needs of Protestant students, but not those of Catholics, who "would want a chapel wherein the Blessed Sacrament is always present".[38] A month later, likely at Killian's request, Baker produced a 5-page memo outlining what he felt were the needs of the campus in a "Chapel Auditorium", with one section seating 1000 people and serving as a kind of meeting house, and another section seating 75 and serving as a chapel. Combined, the building would serve, like the village church or cathedral, as an "essential instrument[s] of community life, of cooperative living, of a well-ordered social structure".[39] This memorandum would

serve as the baseline for the ensuing project; however, the vision it promoted would be significantly transformed in terms of both architecture and the conception of religion that would eventually be articulated in the completed building.

By 1950, Killian had secured the financial sponsorship of the Kresge Foundation, created by Sebastian Kresge, the founder of Kresge stores (and later Kmart) and a devout Presbyterian.[40] A search for an innovative architect led to Eero Saarinen, and Killian's 1950 description of the desired buildings, written in preparation for Saarinen's first visit to campus, offered a detailed understanding of the goals of the project that drew significantly from Baker's memorandum. Killian explained that both buildings would be used for religious purposes although the auditorium, characterized as the "meetinghouse", would also serve secular gatherings. As Killian described it, "the general feeling of the hall should be so noncommittal that it could serve for a religious convocation as well as a Tech Show."[41] In contrast, the small "devotional chapel", should be "reverential in feeling" and "definitively spiritual in intent".[42] The style of the smaller building should have the "aspiring quality" of traditional religious architecture but without using traditional features. He hoped for "romantic beauty" in the chapel, "not austerity and coldness" but "richness and beauty", words that harkened back to Bosworth's influence.[43]

Despite these more inclusive articulations, neither Killian nor Baker, at this early stage, conceived of the chapel as what we now would call "interfaith", that is, as serving a range of religious traditions equally. They conceived of the space as Christian, but all would feel comfortable in it—a notion that if retained would have likely run aground on the shoals of exclusivism just as it had with the Chapel of the Four Chaplains and the Brandeis Chapel. However, the trajectory of the developing ideas articulated in the extant materials suggests an intermingling of three conceptions of religion—religion as private, personal reflection; Protestant latitudinarianism; and tri-faith equality and "non-denominational" cooperation—all struggling to move beyond religious exclusivism and toward a more inclusivist understanding of religious life.[44] But it would take time for such an inclusivist understanding to develop. In fact, additional specifications for the buildings, apparently written by Baker in July 1950, were explicitly Christian and closely aligned with the tri-faith conception: the auditorium should have an altar table, moveable pulpit, reredos, choir stalls, and five pulpit chairs. The "related chapel", conceived as connected to the auditorium but with a separate entrance, would be outfitted with an altar table, lectern, reading desk, and two pulpit chairs. The specifications further noted that "Although any religious symbolism in the architecture or design of the chapel should be predominantly Christian, ideally the religious symbolism should be such that the chapel could be used for religious services for men and women of all faiths."[45] In the dean's view, and likely that of other campus leaders and the donor, the Christian context was paramount. Indeed, a representative of the American Unitarian Association, Fredrick M. Eliot, was brought in as a consultant on the "religious aspects of the new Auditorium-Chapel".[46] Other religious groups that might use the chapel were not named, apart from a passing mention that the on-campus Masonic chapter had expressed interest in using the building.[47] Moreover, in Baker's view, the individual to be hired as the Dean of the Chapel Auditorium "should not be strongly sectarian" but it was important that "he should, of course, be a Christian Protestant".[48]

Thus, the understanding of "religion" implicit in the initial specifications was quite similar to that informing the previous two efforts in Philadelphia and at Brandeis to create a tri-faith, interfaith space. A latitudinarian host would provide a welcoming tent for all comers. But as we have seen in the earlier examples, those potential "comers" balked at this vision, recognizing the disparities of belonging that were implicit in the invitation to use the building as guests and in the very *sethos* of the buildings. Had MIT continued along this path, a similar fate would have likely awaited. Ultimately, however, a new understanding of "religion"—of what "religion" is—emerged and resulted in a successful interfaith chapel.

### 3.3. Phenomenology as the Shared Ground for Multifaith Space

The sticky conundrum that plagued the earlier efforts—achieving equality among all religious traditions within a sectarian context—would only be solved through a radical reconceiving of the concept of "religion" itself and the concomitant expression of this new conception in the design and construction of a building. This radical new formation of thought and space occurred initially at MIT and was later taken up throughout the country. At MIT it was the newly selected architect, Eero Saarinen, fresh off the failed effort to create a building usable by Protestants, Catholics, and Jews at Brandeis, who articulated, initially in text and later in architecture, a universalized idea of human experience that linked all religions and which made possible equality among diverse religious users of a single building. This shift in the understanding of "religion" itself, or what aspect was most important about it—and therefore how it should be accommodated—was brought about by the supplanting of the earlier focus on religious traditions (Protestant, Catholic, Jewish) and their practices with a new emphasis on the religious experience of individuals, that is, phenomenology, blended with the earlier idea of personal religion.[49] As we will see, the gradual development of this alternative understanding of religion, which would come to be expressed in the MIT chapel, can be traced chronologically, from the initial charge given to Saarinen to by the MIT leadership to create a Protestant Christian space in which others would feel comfortable, through Saarinen's conceptualization of religious experience as inspiration for the building design, into the resulting building itself, and culminating in the rhetorical integration of the building and its purpose with the secular mission of the institution.

The first step in the process occurred, as described above, in early October 1950 when Killian introduced the project to Saarinen, but this was no ordinary meeting, for it occurred just a month after both Dean Baker and Matthew Nowicki perished in the crash of TWA flight 903 near Cairo, Egypt on August 31.[50] The untimely deaths of these two men (Baker was 50 years of age, Nowicki was 40), likely hung over the meeting and certainly over Saarinen as he developed the design for the chapel. In the next several months, Saarinen shifted the problem of chapel building away from the traditional context of worship practice and repositioned it in the experiential realm of phenomenology—the individual encounter with transcendence—thus reimagining the function of the religious building. Critically, the concepts of "transcendence" and "spirituality" were not linked to specific religions. For Saarinen, they were experiential and available to all. This conception was far more mystical than Baker and Killian's thought, closer, in fact, to Bosworth's early concern for freethinkers, but it was also tinged with a concern for mortality, and it focused on the psychological components of religion.

Expressing his understanding of the problem in an undated letter to his frequent correspondent, Astrid Sempe, Saarinen wrote about his new design for the MIT chapel:

> It is a building conceived entirely—or almost entirely thru [sic] a psychological approach. –What mood [illegible] is the best to inspire in man religious thoughts? (In the individual man not in groups like weddings etc. because they create their own environment)—Should a chapel be all glass within a [court] so that you can see the sky dome? This would give you the greatest feeling of expansiveness— perhaps. Should it be all dark and enclosed and perhaps padded like a womb (*livmoder*)? Should it somehow recall to you the feeling of death toomblike [sic]? Should it give you the feeling you get if you sit on top of a mountaintop at night? Should it be underground—above ground.—In otherwords should it give you the feeling of loneliness—or security—or doom—or gregariousness or what the hell should it do.—Conclusion—darkness but not absolute darkness—Mountain top combined with the funereal.[51]

This was a distinctly experiential conception of religion, vastly different from the Protestant Christian, even non-denominational, understanding that focused on accommodating worship services, that Killian and Baker had outlined.[52] Saarinen went on in the



letter to explore the role of the natural environment in conjuring the psychological state he wished to create. Using his own personal experience near a small town in Greece as an example, he wrote:

> The town is built on a hillside and overlooks a beautiful valley which goes all the way to Sparta. Opposite the town there is a smaller hill on which they had the cemitary [sic]. It was moonlight and we climbed up to the top of the mountain on the side which Andrizzena [Andritsaina] was located and sat on stones and looked over the valley—the whole sky dome was over us with every star and in front way down was the cemetery [sic]—I felt that was the right setting. If one could create such a mood or feeling in a chapel that would be the best.[53]

Describing the developing building, he wrote "the chapel will have an almost medieval feeling—almost Romanesque—looks a bit like Castello St. Angelo (Hadrian's Tomb) in Rome".[54]

This conception of a tomb-like space, emphasizing the themes of life, death, and mortality, echoed Bosworth's discussion of the "imponderables" from years earlier. Such concerns unite all humanity and should be reflected on by individuals—alone in an isolated space, apart from the distractions of the everyday world. This conception of religion, not as the communal worship practices organized by specific traditions but as the personal experiences of individuals contending with their own consciences and their own struggle to understand the infinite, freed both Saarinen and the MIT leadership from the conundrums of dealing with relationships among religious groups by moving "religion" into the realm of universal human experience. Although the letter is not dated, it likely was written in 1952, for he mentioned that the model for the new chapel was complete.[55] In January 1953, Saarinen's phenomenological descriptions used in the letter along with photographs of the chapel model were published in an article that ended on his aspiration: "Civilizations of the past seem to have placed a greater almost spiritual value on architecture. . .is it not possible that architecture may, some day, play this higher role again?" (Saarinen Challenges the Rectangle 1953, p. 133).

The themes of personal religion, individual experience, spirit, and spirituality would subsequently be repeated in other writings by Saarinen and adopted by the MIT leadership as the project went forward, and the previous language regarding interdenominational or non-sectarian usage all but gave way to this conception of a universally shared experience and, in the MIT version, ethical values. Saarinen's experiential approach to fostering religious feeling through architecture was hardly a new idea. But its identification as the conscious impetus for the unique design of the building coupled with the depiction of experience as not necessarily connected with a specific religious tradition but relevant to all, provided a way to reimagine religious space, not as historically moored to earlier religious forms and spaces but as fundamentally creative, inclusive, personal, and private. Creating spatial effects based on environmental phenomena in order to tap into human psychology, Saarinen's expressionistic design for the chapel played a key role in achieving the goal of a space accommodating practitioners of all religions and none but also in shifting the understanding of "religion" away from traditions and denominations.[56]

The physical features of the chapel articulated the new focus on individual spiritual experience, emphasizing what religious studies scholar Wilfred Cantwell Smith would refer to a few years later as the relationship between man and the transcendent. (Smith 1962, pp. 154–57). Specifically, the building's physical isolation on the campus and fortress-like separation from the outside world signaled the individual, personal aspect of the experience, which Nowicki had captured in a preliminary drawing for the initial Brandeis proposal and which Saarinen redrew for MIT (Figures 5 and 6). The cylindrical form of the building and its exterior blind arches alluded not only to Roman tombs, as Saarinen noted, but also to Christian baptistries such as those in Ravenna and Pisa, suggesting both life and death (Figure 7). The low lighting of the interior suggested what contemporaneous poet, Dylan Thomas called a "close and holy darkness." (Thomas 1954, p. 31). What little light there was in the building was mostly natural light from the oculus and indirect lighting around

the perimeter just above floor level where a series of windows overhanging the exterior moat allowed outdoor light to filter up into the space, a strategy replicating Saarinen's description of his experience in Greece.[57] The interior walls of the chapel undulated, as the early drawings by Nowicki specified, the bricks were laid with uneven projecting bricks to create a highly textured surface, and the venting panels consisted of bricks laid with positive and negative spaces—all suggestive of Saarinen's description of the flickering light he experienced in the cave of the Blue Grotto on the island of Capri in Italy and creating the feeling of womblike containment (Figures 8 and 9).[58]

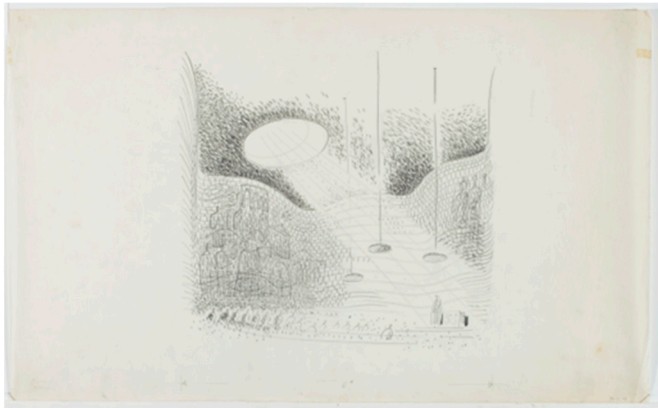

**Figure 5.** Sketch of individual in the Brandeis Chapel by Matthew Nowicki, 1950. Courtesy of Cranbrook Archives, Cranbrook Center for Collections and Research.

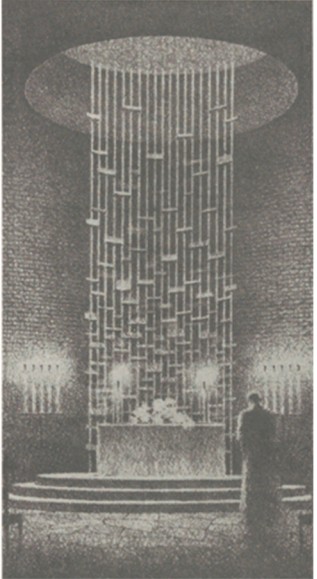

**Figure 6.** Sketch by Eero Saarinen of an individual in the MIT Chapel. Dedication Program, 1955. MIT Planning Office Records, Series 4; Academic Real Estate, Chapel, Building W15 (photographs); AC-0205, Box 4; Department of Distinctive Collections, MIT Libraries, Cambridge, Massachusetts.

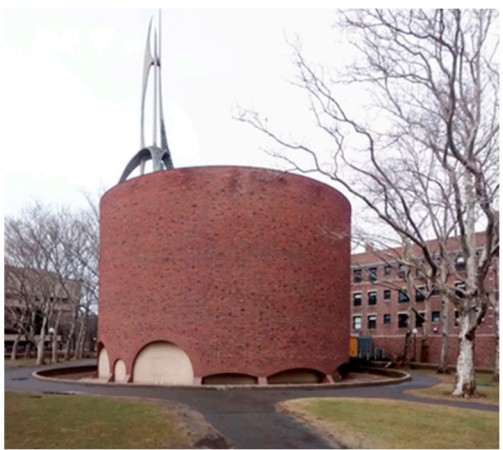

**Figure 7.** MIT Chapel Exterior. Photo courtesy Gunnar Klack, CC BY-SA 4.0 <https://creativecommons.org/licenses/by-sa/4.0>, via Wikimedia Commons, https://commons.wikimedia.org/wiki/File:MIT-Chapel-Eero-Saarinen-Massachusetts-Institute-of-Technology-Cambridge-Massachusetts-2014-04-01.jpg (accessed on 27 January 2024).

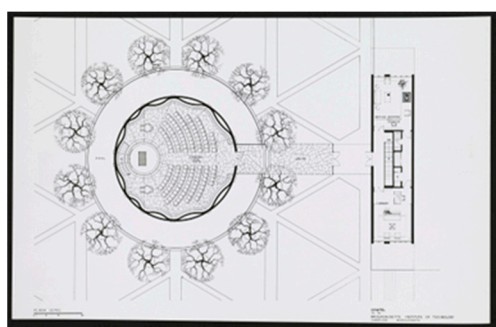

**Figure 8.** Eero Saarinen & Associates, Kresge Chapel plan, 1950. Library of Congress https://www.loc.gov/item/2008680851 (accessed on 28 January 2024).

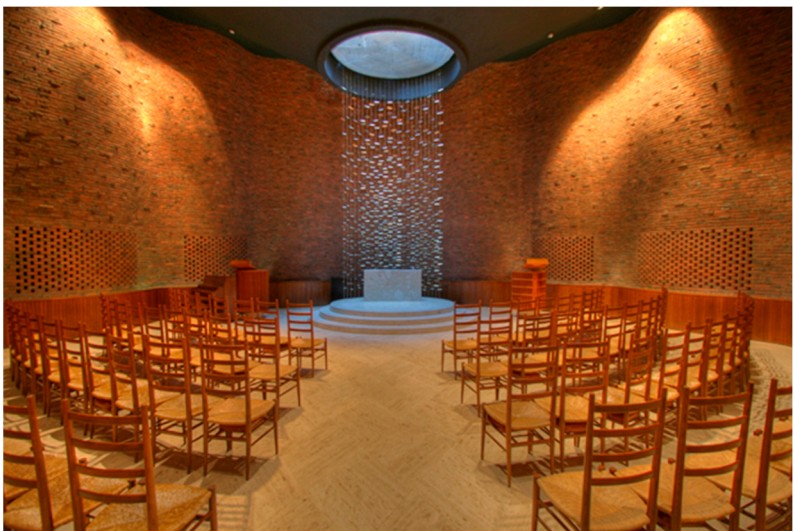

**Figure 9.** MIT Chapel Interior. Photo courtesy Madcoverboy at English Wikipedia, CC BY-SA 3.0 <https://creativecommons.org/licenses/by-sa/3.0>, via Wikimedia Commons, https://en.wikipedia.org/wiki/File:MIT_Chapel_Interior.jpg (accessed on 27 January 2024).

In addition to these elements signaling nature and personal experience, the building's few sculptural elements also gestured toward an inclusivist (rather than an exclusivist)

understanding of religion. The raised marble altar provided a focus that pointed to the Christian use of the building, but artist Harry Bertoia's sculpture of shimmering metal strips cascading from the oculus to the floor behind the altar signaled no specific religious tradition. In fact, it echoed Bosworth's earlier suggestion for an electrified altarpiece that would be acceptable to freethinkers. The glittering sculpture/mobile directed one's eye to the oculus above and onto the heavens beyond. In this dimly lit, enveloping interior space, the individual, contemplative and secluded, might privately experience something revelatory, however that "something" might be conceived[59] (Figure 9).

A nod to the tri-faith notion of religion appeared on the exterior of the building in the form of Theodore Roszack's three-legged finial on the roof, but Roszack also drew inspiration from other religions and offered additional mystical contexts. In his words, "it was of importance for me—that the design of the chapel in 'plain view' is round, emerging from a 'baptismal fount' (the most) symbolizing the spiritual source of intuitive experience. In gnostic terms, the circle or 'Mandala' becomes the 'magic circle' that designated 'holiness' and is the symbol of transcendent unity as 'Yang and Yin' is to the Star of David."[60] The inscription on the building similarly left the door open to a diversity of religious perspectives: "This building gives embodiment to the responsibility of the Massachusetts Institute of Technology to maintain an atmosphere of religious freedom where-in students may deepen their understanding of their own spiritual heritage, freely pursue their own religious interests, and worship god in their own way".[61]

As the building took shape, MIT's descriptions changed to reflect Saarinen's phenomenological vision of personal religion. As early as March 1952, building committee chair Richard Kimball explained that Saarinen, ". . .has succeeded in interpreting the deep spiritual qualities which should contribute greatly in carrying out the program to develop 'the whole man' at the Institute" and that the chapel ". . ..will be available to all creeds among our student body who wish to make use of this facility to express their individual faiths in a quiet atmosphere created for more personal devotion".[62] By 1954, this understanding of the building was asserted in Killian's presidential address, which emphasized a connection between the creativity of the architectural design and MIT's commitment to scientific and spiritual creativity, uniting the religious and secular within the institution.[63] As he noted repeatedly both the building and the institution "encouraged a creative approach to matters of the spirit". He wrote,

> An institution which embraces general as well as professional education must give attention to man's spiritual life—to the place of religion in man's history, in contemporary society and in the life of the individual. It also must encourage an understanding of those postulates which underlie our society's concept of virtue—the unifying ideals and standards, the moral and ethical beliefs which men in general agree upon but reach by diverse paths of faith, philosophy or social pressure. . .[young people's] all-round development requires a growth of the spirit as well as the mind.[64]

By the time the building was dedicated in May 1955, the rhetoric had shifted almost entirely away from religious traditions and toward personal religious experience and spiritual life. The above quotation from Killian's Corporation address appeared in the chapel dedication program and in his address at the dedication, in which he emphasized the role of the building as a "devotional chapel" that would advance the Institute's commitment to religious freedom by providing a space where small groups can meet and, as the inscription on the building indicated, worship in their own way. (Killian 1955, p. 402). Religion, as depicted at the dedication, signaled the life of the individual "spirit" along with spiritual or philosophical grounding for ethical principles. Killian identified no specific religious groups, nor did E. N. van Kleffens, Minister of State of the Netherlands, who, in delivering the dedicatory address, stressed the need for humanistically derived values of "divine and human" law as a temper on scientific research. (van Kleffens 1955, pp. 403–4). Quoting from Killian, van Kleffens defined the roles of the chapel as,

> First—to stand as a symbol of the place of spirit in the life of the mind and as a physical statement of the fact that MIT has a right and a responsibility to deal with ideals as well as ideas and to be concerned with the search for virtue while we become proficient in the search for things. Second, to provide ready opportunity for students and other members of our community to worship as they choose, to have on campus a building, beautiful and evocative of reverence and meditation, where those who wish may enter and worship God in their fashion.[65]

The shift in the language used to describe the chapel is significant. This was to be an inclusive space open to all, totally different from "sectarian" spaces of the past, but also different from the tripartite Protestant/Catholic/Jewish conception of both the Chapel of the Four Chaplains and the initial Brandeis conception.[66] Only the institutional MIT press release referred to the tri-faith context saying, it was "one of the few [buildings] in the country to be used for regular services by Catholics, Protestants, and Jews alike", but even it quickly moved on to frame the building as "one of the most extraordinary religious buildings of our time... designed for quiet retreat."[67] Here was a new religious purpose—addressing the individual needs of those seeking a personal experience—requiring a new religious architecture.

Both the inclusivist and the experiential goals were soon realized in the building's usage. In the weeks following the opening of the building, dedicatory services were held by several groups: Protestants (May), Episcopalians (September), and Greek Orthodox (October).[68] On October 7, the Archbishop of Boston, Richard J. Cushing, celebrated Mass in the building.[69] Rosh Hashanah services were held in the chapel in September.[70] After these festivities passed, weekly schedules drawn up by the Associate Dean of Students, Robert Holden, regulated the multi-religious and individual use of the building. In March 1956, for instance, the building schedule included weekly Vedanta (Hindu), Christian Science, and Greek Orthodox services along with Shabbat, Episcopal Holy Communion, Legion of Mary, and Compline (Christian) services. Catholic Mass was celebrated daily at 7:55 A.M., followed by Morning Prayers, after which the building was reserved for private meditation.[71] A year later Holden estimated that during an average day, some 40–50 people used the chapel for private meditation.[72] Killian, reflecting on the chapel years later, praised Saarinen for "skillfully fulfill[ing]" the requirement that the building "not express commitment to any particular tradition in its design" and "creating a meeting place adaptable to almost any kind of religious service, funeral, wedding, or meditation." (Killian 1985, p. 239).

Thus, this chapel was successful in serving a variety of religious groups and religious purposes at MIT. But simply providing a useful or amenable space was not the extent of the building's contribution to the campus, for in sharing spaces, diverse groups were becoming more engaged with one another. Only a month after the dedication, Holden noted that the chapel had been witness to "more frequent cooperative relationships" between the Protestant Christian organization, the Technology Christian Associate, Hillel, the Catholic Club, and other groups and that cooperation indicated a new direction in student affairs.[73] National interest in the architecture of the chapel soon grew, as did interest in its individual and multifaith use. Requests for information about the chapel poured in. The YMCA General Secretary at Texas A&M University, for instance, requested particulars on the chapel's use and success as an "interfaith" chapel, and particularly the extent of its use by individuals for meditation. Requests for the inscription and information on how scheduling was handled were frequent.[74] Given the level of interest among chaplains from a host of other institutions, it would not be long before the MIT chapel would become a model for multifaith spaces nationwide, both on college campuses and in other public locations (Kilde 2017).

By emphasizing the experiential aspect of religion within a secular, scientific campus, Saarinen and MIT succeeded in creating a building that would serve not only personal meditation but also the variety of groups and services that latitudinarians had sought for over a decade. From the initial, Protestant-informed decision through the designer and the design process and on to the rhetorical interpretation of the building and its eventual

use, we can see the graduate shift in the understanding of what religion is, what elements need spatial accommodation, and how spaces might be designed to do so. For MIT and Saarinen, the key to the successful multifaith space was de-prioritizing the accommodation of worship practice—for men in groups, Saarinen noted, "create their own environment"—and fostering the individual, psychological experience arising from spiritual reflection. United by the context of this universalist understanding of religious experience, all groups could feel equally at home in the building.[75]

With these three examples, then—the Chapel of the Four Chaplains, the Brandeis University chapels, and the MIT Chapel—we can see the gradual movement, conceptual and architectural, toward a viable space serving many religions. Achieving that goal, however, required overcoming long-held exclusivist understandings of religion that precluded spatial cooperation and moving beyond the hope that the goodwill and tolerance of a host tradition would prove sufficient for spatial sharing. As we have seen, the creation of a multi-religious shared space required shifting the foundational conception of "religion" itself away from the traditional understanding of the distinctiveness of religious traditions and their worship practices toward an inclusivist, phenomenological conception of religion as a psychological spiritual experience shared across traditions. In this way, the development of the multifaith chapel not only marks a revolution in the history of religious space but also a significant cultural shift in how "religion" itself was understood.

**Funding:** This research was funded by the University of Minnesota.

**Data Availability Statement:** No new data were created or analyzed in this study. Data sharing is not applicable to this article.

**Conflicts of Interest:** The author declares no conflict of interest.

## Notes

1.  Throughout this essay, I will use the latter two terms interchangeably, generally following the nomenclature of the period and context under examination. I use the term "non-denominational" only when quoting from original sources in order to avoid confusion given its roots in Christian denominationalism and its current use to identify many evangelical congregations that eschew affiliation with historical Protestant groups.

2.  Scholarship on the development of the concept of "religion" is vast. See, for instance, (Asad 1993); (Smith 1998); (Masuzawa 2005); and (McCutcheon and Arnall 2013). On religious exclusivism and inclusivism, see (Laine 2014); (Moser 2010); (Eck 2003). Contemporary discussion of exclusivism and pluralism in the field of philosophy of religion is vast. See, for instance, (Meeker 2006); (Hick 2006); (King 2008).

3.  On twentieth century anti-Catholicism, see (Massa 2001); (McGreevy 1997). A useful overview of scholarly trends in the historical study of antisemitism in the US is (Tevis 2021).

4.  On military chaplaincies, see Wendy Cadge's (2022) groundbreaking study.

5.  On the slow growth of cooperation across traditions and within Christianity (ecumenism), along with a definitive discussion of the Four Chaplains story and the development of the tri-faith model, see (Schultz 2011). One of the earliest stories about the heroic episode appeared as (Four Chaplains 1944).

6.  Sinatra and Davis were Catholic, Lawford Protestant. Davis would become involved with Judaism in the mid-1950s and convert in 1961.

7.  Footage of the rotating altar can be seen here at "Chapel of the Four Chaplains 1952," YouTube, https://www.youtube.com/watch?v=IEiO8iX8keE, accessed 15 January 2024.

8.  (Schultz 2011). Schultz (pp. 4–6) quotes from the 12 September 1960, speech and Q &A session in which JFK publicly refuted this accusation. (Kennedy 1960).

9.  On the anti-Catholicism of "cosmopolitan intellectuals" see (McGreevy 1997, pp. 98–100). See also, (Massa 2001, p. 549).

10. Quoted in (Schultz 2011, p. 6). Question and Answer Period Following Speech of Senator John F. Kennedy (1960).

11. John Dewey's study of Polish Catholics in Philadelphia concluded that the community was, in McGreevey's words, "in thrall to anti-democratic interests." (McGreevy 1997, p. 105). Massa notes that sociologist Paul Blanchard, who assisted with Dewey's study, went on to express a "militantly hostile" view of this so-called Catholic anti-democratic culture, denouncing its threat to the nation within the Cold War context. (Massa 2001, p. 554).

12. When the chapel opened, Monsignor Thomas McCarthy of the National Catholic Welfare Conference noted in an interview with *Time Magazine* that there was no "official Catholic altar" in the chapel and "no Catholic representative was present," because

"canon law forbids joint worship". See (Four Chaplains 1951). Only in 1960 would a Catholic layman, Dr. Shayne MacCarthy, appear on the platform in the Four Chaplains Chapel, and then only under strict rules that the event not be characterized as religious and that none of the officiating clergy wore clerical garb. See (Catholic Layman Condescends to Appear at Interfaith Event 1960). Currently, only Baptist and non-denominational Protestant services are held in the chapel, which was relocated in 2001 to the former Navy Chapel in Philadelphia.

[13] Pre-cursors of inter-faith advocacy included groups such as the National Conference of Christians and Jews that advocated tolerance for all groups and sponsored traveling Tolerance Trios to hold events encouraging cooperation. See (Schultz 2011, pp. 26–29, 35–42).

[14] (Saarinen, Saarinen, and Associates n.d.) Available in the Cranbrook Archives (Saarinen Family Papers, Box 5, Folder 6) and the Yale Archives (MS 593, TN 243480, Box 697, Folder 198). See also, (Bernstein 1999).

[15] Brandeis Reveals Plan (1950), vol. 1. A few drawings from the Master Plan are also available in (Schafer 1973, pp. 37–42).

[16] Brandeis President Abram L. Sachar later commented that in the plan with three buildings, the groups are situated so that "backs will never be turned toward any one religious group." (Plan Simultaneous Building of Three Campus Chapels 1953, p. 1).

[17] In January 1953, Sachar announced that a donation by Dr. David D. Berlin would fund the design and construction of a separate Jewish chapel on campus, which would offer hospitality "to any group which wants to use it." (New Chapel Will be Jewish Sponsored—And Sectarian 1953).

[18] (S.C. Protests Chapel Plan 1953) and (Asks University to Consult with Catholic Authorities 1953, p. 1) and (Letters to the Editor: The Chapel Question 1953, p. 2). On the founding of Brandeis, see (Jick 1995, pp. 103–4).

[19] Additional resentments likely fueled Feeney's animosity, include Cushing's censure of him in 1949, his subsequent dismissal from the Society of Jesuits, and ultimate excommunication in 1953. On Feeney's activities in Boston, see (Savadove 1951). See also, (Levine and Harmon 1992, p. 38).

[20] Feldberg's analysis is muddled. Echoing earlier interpretations, including President Sachar's, he argues that Feeney's objections to Jews were theological, "rather than economic or racial" (p. 112) even though he notes Feeney's accusations that Jews were communists and deceitful and that they killed priests and raped nuns—all antisemitic tropes.

[21] Discussion of the growth of free-thinking and atheism on campuses can be found in Leigh Eric Schmidt (2012); Leigh Eric Schmidt (2016); and Robert S. Ellwood (1997).

[22] Letter from William Welles Bosworth to Irwin [sic] H. Schell, 30 November 1938, AC-0004, Box 33, Folder 9, Office of the President, records of Karl Taylor Compton and James Rhyne Killian, Department of Distinctive Collections, MIT Libraries, Cambridge, Massachusetts.

[23] Bosworth to Schell, 30 November 1938. In a handwritten note to Schell from 1954, Bosworth reminisces about "that chapel for meditation we planned". William Welles Bosworth to Professor Irwin H. Schell, J[anuary] 30, 1954. Fawcett papers, Department of Distinctive Collections, MIT Libraries, Cambridge, Massachusetts.

[24] Bosworth to Schell, 30 November 1938, p. 2.

[25] Bosworth to Schell, 30 November 1938, pp. 2–3.

[26] On auditorium churches, see Jeanne Halgren Kilde (2002).

[27] Bosworth to Schell, 30 November 1938, p. 4.

[28] See note 27 above.

[29] See note 27 above.

[30] Margaret M. Grubiak discusses Bosworth's plan in (Grubiak 2014, pp. 99–102).

[31] The usual reasons, other institutional priorities and funding concerns, stood in the way of the construction of the building. In 1943, then-president Karl Compton rejected building the chapel as "a luxury which should not be sought unless it should come as a gift, either spontaneous or from a donor who would be definitely more enthused over this than some alternative project." Karl Compton to William Welles Bosworth, 13 July 1943, Office of the President, records of Karl Taylor Compton and James Rhyne Killian, AC-0004, Box 33, Folder 9, Department of Distinctive Collections, MIT Libraries, Cambridge, Massachusetts.

[32] The term "big science" is generally attributed to Alvin M. Weinberg (1961, pp. 161–64), who critiqued the rise of the military-industrial complex in his article.

[33] On the relationship between MIT's military research, the development of it humanities curriculum, and its religious rhetoric, see also (Martin 2013, pp. 83–86).

[34] William W. Bosworth to Karl Compton, 7 January 1945. Office of the President, Records of Karl Taylor Compton and James Rhyne Killian, AC--0004, Box 33, Folder 9, Department of Distinctive Collections, MIT Libraries, Cambridge, Massachusetts. On Colepaugh, see (A Connecticut Nazi Spy Has a Change of Heart 2020). The "Dr. Butler" referred to here is most likely Nicholas Murray Butler, then president of Columbia University.

[35] MIT was not alone in this. On the effort to integrate ethics into science-oriented institutions, see (Grubiak 2007, pp. 1–14).

[36] See, for instance, Ellwood's discussion of Harvard President Nathan Marsh Pusey, an Episcopalian who emphasized personal religion and tolerance (Ellwood 1997, pp. 195–96).

[37] All quotations from Everett Moore Baker to Mr. [James] Killian, 9 February 1948. MIT Archives, AC4 Box 131, Folder 5.

[38] John H. O'Neill, Jr. to Mr. [J. R.] Killian, 9 February 1948. Office of the President, Records of Karl Taylor Compton and James Rhyne Killian, AC4, Box 131, Folder 5, Department of Distinctive Collections, MIT Libraries, Cambridge, Massachusetts.

[39] Everett Moore Baker to Dr. Killian, 12 December 1949, Office of the President, Records of Karl Taylor Compton and James Rhyne Killian, AC4, Box 131, Folder 5, Department of Distinctive Collections, MIT Libraries, Cambridge, Massachusetts.

[40] On the MIT appeal to the Kresge Foundation, see Joseph M. Siry (2018, p. 278). The Kresge Foundation had also contributed to the Chapel of the Four Chaplains in the late 1940s.

[41] James Killian to R. M. Kimball, 3 October 1950, page 1, Office of the President, Records of Karl Taylor Compton and James Rhyne Killian, AC4 Box 131, Folder 5, Department of Distinctive Collections, MIT Libraries, Cambridge, Massachusetts.

[42] James Killian to R. M. Kimball, 3 October 1950, page 2.

[43] See note 42 above.

[44] Siry's (2018) informative article emphasizes the latter two concepts, focusing on the institutional embrace of a "non-denominational" ethos rooted in the liberal Protestantism of the institute's leaders. My interest, in contrast, lies in the relationship among all three conceptions and just what the terms meant at the time, that is, in the politics of their use as illustrative of the beginnings of a significant shift in how Americans understood "religion". The fluidity of the terms used to characterize the need and uses of the new space suggests the unstable, transitional character of the conceptions of religion and engagement among religious groups behind them. Architectural historian Reinhold Martin (2013) is also interested in the discursive strategies and contexts of the MIT leaders during this period. Focusing on the relationship between the development of the chapel and the development of the Institute's School of Humanities and Social Sciences, Martin emphasizes the intervocality of the two buildings, which in his view express religious (the chapel) and secular (the auditorium) concerns.

[45] "Specifications for Auditorium-Chapel for M.I.T.", typescript, 17 July 1950, page 3. A handwritten note at the top of the first page indicates the document was "written by Baker". Office of the President, Records of Karl Taylor Compton and James Rhyne Killian, AC4, Box 131, Folder 5. Department of Distinctive Collections, MIT Libraries, Cambridge, Massachusetts.

[46] Frederick M. Eliot to [R.M.] Kimball, 22 March 1951; R.M Kimball to James R. Killian, Jr, 27 March 1951, MIT Office of the President, Records of Karl Taylor Compton and James Rhyne Killian, AC4, Box 131, Folder 5, Department of Distinctive Collections, MIT Libraries, Cambridge, Massachusetts.

[47] "Specifications for Auditorium-Chapel for M.I.T.", typescript, 17 July 1950, page 3.

[48] Everett Moore Baker to Dr. [James] Killian, 6 July 1950, Office of the President, Records of Karl Taylor Compton and James Rhyne Killian, AC4, Box 131, Folder 5, Department of Distinctive Collections, MIT Libraries, Cambridge, Massachusetts.

[49] The term *phenomenology* has a chequered reputation in the field of religious studies. As a method of inquiry that seeks to identify and understand particular phenomena—specifically, experiences and/or the structures that make experiences possible—that are evident across (all) religions, it became popular among scholars of religion in the early-to-mid-twentieth century, the same period under examination in this article. See Gerardus van der Leeuw (1938); and Mircea Eliade (1961). The conception of universally shared religious experience posed at the time also resonated with religious practitioners outside the academy who were seeking cooperation among religious groups and found experiential similarities a foundation for such. It remains important for such practitioners to date. For religious studies scholars, however, phenomenology as a method of inquiry generally lost its usefulness by the late twentieth century, as its dehistoricized understanding of religion as a universal phenomenon was supplanted by cultural and functional models of religion. In this article, I use the term not as a method of inquiry but to identify the specific, historical understanding of the concept of religion that emphasizes experience. Useful overviews of the varied uses of the term phenomenology and its rise and fall among scholars are available in Studestill (2000, pp. 177–94) (which is sympathetic to phenomenology as a method); and Hughes and McCutcheon (2022, pp. 193–98) (which is not).

[50] Baker was returning from a conference in Bombay, and Nowicki, then serving as the head of the Architecture Department at North Carolina State College, was traveling for meetings related to his recent commission to design the new city of Chandigarh in northern India. The flight had originated in Bombay, stopped over in Cairo, and crashed shortly after take-off from Rome. The MIT campus grieved the loss of Baker, eventually renaming a new dormitory in his honor and establishing a memorial endowment in his name. Nowicki's Chandigarh project was taken over by Le Corbusier. It is not known whether the two men knew one another. See "Everett Moore Baker papers, Biographical Note," *MIT Archives Space*, MIT Libraries, https://archivesspace.mit.edu/repositories/2/resources/598 (accessed on 15 January 2024) and "(Wikipedia 2023), https://en.wikipedia.org/wiki/TWA_Flight_903 (accessed on 15 January 2024). It should also be noted that Saarinen's father, Eliel, passed away on July 1 that same summer.

[51] Eero Saarinen to Astrid Sempe, n.d. Cranbrook Archives, Astrid Sempe Collection of Eero Saarinen Correspondence, Box 3 Correspondence, 1952. Folder 5. Correspondence, N.D., Cranbrook Archives, Cranbrook Center for Collections and Research, Bloomfield Hills, Michigan.

[52] The experiential understanding of religion has deep roots in pietistic Lutheran thought, which Saarinen was likely familiar with given that his grandfather was a Lutheran minister. Surviving papers, however, make no mention of his specific ideas about Lutheranism. (Kilde 2014).

53     Eero Saarinen to Astrid Sempe, n.d.

54     See note 53 above.

55     That Saarinen was thinking about the spiritual possibilities of modern architecture at this time is also apparent in an essay he presented to the American Institute of Architects in October 1952, in which he identified "the greater spiritual meaning of architecture to our civilization" as one of the three fundamental problems or challenges facing architects at the time. See Eero Saarinen (1952, p. 245).

56     It should be noted that in "Saarinen Challenges the Rectangle", the design of the chapel is credited to Saarinen and Bruce Adams.

57     The oculus is, of course, a nod to the Pantheon in Rome, an ancient, "multifaith" building. The innovative perimeter lighting proved to be insufficient. Complaints leading to additional artificial lighting occurred soon after the chapel was opened. During my research, I learned from the Reverend John Wuestneck, a member of the institute's chaplaincy, that the lighting was never particularly functional, in part because the trees surrounding the building shed their leaves into the moat, blocking the sunlight. Further, initially, there were no screens or air conditioning, so the building was stifling in the summer. Author's personal conversation with John Wuestneck, 30 May 2017, Cambridge, MA.

58     The Cold War theme of containment is also relevant here. See Elaine May (1988). Nowicki's influence on Saarinen was significant. See Tyler Sprague (2010).

59     For an alternative interpretation of the spiritual aspects of the chapel and its multifaith success, see Joseph Siry, who traces the roots of the chapel to the liberal Protestantism of the MIT leaders and argues that the "symbolic familiarity" of the Christian features provided a "bridge" to the "non-denominational ideal" and "accessibility . . .for all faiths" (275). My argument pushes this point somewhat, agreeing that liberal Protestant ideals were indeed at play, particularly in 1950–1952, but asserting that they became superseded by the metaphysical and phenomenological emphasis that Saarinen brought to the project and which the MIT leadership adopted by 1953, and that the significance of this new emphasis and language lay in its redefining of religious space and religion itself. In any case, we agree, as Siry concludes, that the space "transcended denominational and national cultures" and that the "elemental spiritual effect" was paramount (290).

60     Letter from Theodore Roszack to Mr. Wylie, 24 November 1955, page 2, Office of the Dean for Student Affairs, Records of Associate Dean Robert Holden, Box 6, Folder "Kresge Chapel, 1/2," Department of Distinctive Collections, MIT Libraries, Cambridge, Massachusetts.

61     While this article engages with the MIT chapel from the perspective of the field of religious studies, scholarship from the field of architectural criticism and history may be of interest to readers. In addition to sources cited herein—particularly Grubiak (2014), Martin (2013), Siry (2018), and Sprague (2010)—recent architectural treatments of the chapel can be found in Jayne Merkel (2005); José Ignacio MartÍnez Fernández (2015); and Jennifer Komar Olivarez (2006).

62     R. M Kimball, "Suggested Remarks on Kresge Auditorium-Chapel for M. B. Dalton's Detroit Speech—31 March 1952," typescript, MIT Office of the President, Records of Karl Taylor Compton and James Rhyne Killian, AC4, Box 131, Folder 5, Department of Distinctive Collections, MIT Libraries, Cambridge, Massachusetts.

63     Writing years later about the chapel in his autobiography, Killian quoted at length from his 1954 President's Report and its emphasis on "[encouraging] a creative approach to matters of the spirit" (Killian 1985, p. 241). Killian's emphasis on creativity may well have also been in response to early criticism of Saarinen's unconventional design.

64     James Killian, Report to the M.I.T. Corporation, October 1954, printed in James Killian, "President's Report Issue, *Massachusetts Institute of Technology Bulletin*, 90, no. 2 (November 1954), 29–32, https://web.mit.edu/src/pres-rep/49-58__Killian/1954.pdf, (the quote appears on p. 29), accessed on 15 January 2024, and in Killian (1985, p. 241).

65     E. N. van Kleffens, "The Dedicatory Address." See also Killian's letter to van Kleffens, which includes this quotation. James Killian to E. N. van Kleffens, 1 April 1955, Records of Karl Taylor Compton and James Rhyne Killian, AC4, Box 131 Folder 8, Department of Distinctive Collections, MIT Libraries, Cambridge, Massachusetts. In this letter, Killian goes on to say that "the chapel may serve to encourage a creative approach to matters of the spirit. . .the institution of science may well be an environment especially favorable to deeper spiritual insights. More important than its practical achievements are the spiritual contributions of science, its emphasis on the importance of truth and the value of brotherhood, and its revelation of the beauty, the order, and the wonder of the universe. Through these contributions it shares with the great faiths opportunities for furthering man's spiritual understanding; and creative minds and spirits, availing themselves of the resources of both science and religion, may advance man's search for virtue and understanding with new vigor and in new ways." See also Grubiak (2007).

66     James Laine defines "inclusivist" as an "intellectual or political approach to religion, which assumes that differing religions can be included within an overarching system" (Laine 2014, p. 6).

67     News Release, From the News Service, MIT, 1 May 1955, MIT News Office, AC0069_195505_037, Department of Distinctive Collections, MIT Libraries, Cambridge, Massachusetts (https://cdn.libraries.mit.edu/dissemination/diponline/AC0069_NewReleases/NewsRelease_1950/AC0069_1955/AC0069_195505_037.pdf) (accessed on 15 January 2024)).

68     Robert J. Holden, "M.I.T. Chapel and Kresge Auditorium," 15 July 1956, typescript. Office of the Dean for Student Affairs, Records of Associate Dean Robert Holden, Box 6, Folder, "Kresge Chapel 2/2," Department of Distinctive Collections, MIT Libraries, Cambridge, Massachusetts.

[69] News Release, From the New Service, MIT, 7 October 1955, MIT News Office, AC0069_195510_005, Department of Distinctive Collections, MIT Libraries, Cambridge, Massachusetts (https://archivesspace.mit.edu/repositories/2/digital_objects/2206, accessed on 30 January 2024). On Cushing's decision, see a series of letters, Office of the President, Records of Karl Taylor Compton and James Rhyne Killian, AC 4, Box 131 Folder 9, Department of Distinctive Collections, MIT Libraries, Cambridge, Massachusetts. See also, Killian (1985, pp. 239–40).

[70] Robert J. Holden to James R. Killian, Jr., 15 September 1955, Office of the Dean for Student Affairs, Records of Associate Dean Robert Holden, Box 6, Folder: "Kresge Chapel 2/2," Department of Distinctive Collections, MIT Libraries, Cambridge, Massachusetts.

[71] Chapel Schedule, 16 March 1956, Office of the Dean for Student Affairs, records of Associate Dean Robert Holden, AC-0115, Box 6, Folder "Kresge Chapel 1/2, Department of Distinctive Collections, MIT Libraries, Cambridge, Massachusetts.

[72] Robert J. Holden to Mr. J. Gordon Gay, 30 January 1957, Office of the Dean for Student Affairs, records of Associate Dean Robert Holden, AC-0115, Box 6, Folder "Kresge Chapel 1/2, Department of Distinctive Collections, MIT Libraries, Cambridge, Massachusetts.

[73] "Annual Report of the General Secretary to the Advisory Board of the Technology Christian Association, 20 June 1955, page 3, Office of the Dean for Student Affairs, Records of Associate Dean Robert Holden, AC-0115, Box 6, Folder "Kresge Chapel 2/2," Department of Distinctive Collections, MIT Libraries, Cambridge, Massachusetts.

[74] See various letters, Office of the Dean for Student Affairs, Records of Associate Dean Robert Holden, AC-0115, Box 6, Folder "Kresge Chapel," Box 6, Folder: "Kresge Chapel 1/2," Department of Distinctive Collections, MIT Libraries, Cambridge, Massachusetts. To respond to the many inquiries more efficiently, Holden wrote a brochure describing the building, its use, and its scheduling. Unfortunately, I did not find a copy of the brochure in the MIT archives.

[75] Saarinen to Sempe, n.d. Notions of religious experience or transcendence as universal among all religions (a concept termed "perennialism"), although frequently embraced on a popular level and by many contemporary interfaith groups, have been criticized by many scholars of religion for their failure to sufficiently acknowledge the cultural foundations and differences among religions. See, for instance, Stephen Prothero (2010). See also, note 2, above. In a somewhat similar vein, scholars of religion have tended to discount popular language regarding spirituality—particularly the "I'm spiritual but not religious" attitude—on the grounds that substituting "spirituality" for "religion" is a purely semantic move. In contrast, those taking spiritual language as critical to understanding religious behavior include scholars such as Leigh E. Schmidt (2012) and Jeffrey Kripal (2006). In this essay, I have attempted to trace the constructed and contingent character of the language and how those changes pointed to substantive re-conceptions of "religion" itself and, in turn, the elements of it that needed spatial accommodation. As such reconsiderations of the concept of religion continue to occur within the field of religious studies, attendant reconsiderations of religious space are also needed. For a somewhat similar call see (Grubiak and Parker 2017, pp. 17–22). Grubiak and Parker point to what they see as the "utopian vision of religious pluralism" shared by those who advocated for interfaith buildings. I see the process as less utopian than one of articulating architecturally ongoing religious developments.

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
