# Peer review of "Creating the Multifaith Chapel, 1938–1955: Architecture and the Changing Understanding of “Religion”"

_religions, doi:10.3390/rel15030275_

Round 1

Reviewer 1 Report

Comments and Suggestions for Authors

This is an excellent essay and will be a great contribution not only to our understanding of the history of multi-faith worship spaces, but also to our understanding of changing conceptions of religion and of the history of twentieth-century religious architecture.

The essay does an excellent job of putting a fairly well-known and celebrated space, the MIT Chapel, in the context of "God-and-country" patriotism, fears of "godless science," and, most importantly, previous attempts to provide for a multi-faith worship space. This context is provided by considering two less widely known efforts to create multi-faith worship spaces, which were nevertheless very well known in the years leading up to the MIT Chapel. The differences between the Four Chaplians and Brandeis chapels and the MIT Chapel clearly shows how the MIT Chapel inaugurated a new way forward to the creation of the, now familiar, multi-faith worship space.

This essay provides an essential chapter in the history of multi-faith worship spaces, showing the importance of moving from the idea of one religion hosting others, and from a tri-faith understanding of religion to a phenomenological one. A common "religious space" was that was appropriate to people of many different religions and of no particular religion.  Any editor should be delighted to include it in their volume. It is very well suited to this special issue.

The essay is extremely well situated in the literature and supported by the archival record. It is easy to read, well structured, and engaging.

The one substantive suggestion I have is to provide a footnote for the key term "phenomenology." The word can have lots of meanings. The sense in which this paper is using it is made clear in lines 648 and 663, namely "the religious experience of individuals." But the use of "phenomenology" as opposed to, say, "mysticism" and differientiating it from a Protestant emphasis on personal experience seems clearly to be evoking the phenomenological approach to religion that was becoming prominent in America around this time (Otto, Eliade, van der Leeuw, et al..) A footnote would be helpful to point interested readers to a work or more that informs the use of the term "phenomenology" in this essay.

There are a few things to be cleaned up in copy editing. Those I noticed included indentation of paragraphs that are block quotes (e.g., ln 689ff), "acheived" instead of "achieve" in line 17, "altar" instead of "atler" in line 264.

Author Response

Thank you very much for this generous review. The summary you've provided reassures me that the significance of my argument is clear.  The suggestion to provide a footnote explaining the concept of phenomenology and its development is very helpful, and I will definitely do so in the revision. I will also make the editing corrections noted.  Thank you again.   

Reviewer 2 Report

Comments and Suggestions for Authors

The essay offers an in-depth reflection on the categories of thought and social practices underlaying the three architectural case-studies, which are explored with great philological rigor on primary textual sources and political contexts.

It is nevertheless clear that the author's - anonymous - field of research is not strictly architectural criticism: this is evident from the absence of a reference note on the numerous researches that, internationally, have dealt with the topic in its general architectural and spatial characters and its periodization on a broad scale, as well as from the absence of references to Saarinen's historiography, which is very rich.

Such historiographical gaps do not detract from the - excellent - value of the essay, but probably a reporting of the literature on the topic (also highlighting its limitations or specializations or lacunae) and on the best-known building would open up to readers from different disciplines scenarios for further study. Discussing spaces, the primary sources are not only the documentary ones (excellently investigated with brilliant results), but also the material ones, which have their own specific exegesis. In any case, congratulations on an excellent essay.

verify layout 508-529, 672-682, 689-695, 804-810

verify line 361

verify inverted commas 711-712 

last paragraph title 878-879? are conclusions missing?

Author Response

Thank you for this helpful review. I appreciate the suggestion that the addition of historiographic information on architectural discussions of interfaith chapels and the MIT chapel would benefit readers across various fields, and I will add a footnote to provide such information.  (The reviewer is correct in the assumption that my field is not architectural criticism.)  I will also make the editing corrections noted. Thank you.